# Digital Online Patient Informed Consent for Anesthesia before Elective Surgery—Recent Practice in Europe

**DOI:** 10.3390/healthcare11131942

**Published:** 2023-07-05

**Authors:** Claudia Neumann, Nadine Straßberger-Nerschbach, Achilles Delis, Johannes Kamp, Alexandra Görtzen-Patin, Dishalen Cudian, Andreas Fleischer, Götz Wietasch, Mark Coburn, Ehrenfried Schindler, Grigorij Schleifer, Maria Wittmann

**Affiliations:** 1Department of Anesthesiology and Intensive Care Medicine, University Hospital, 53127 Bonn, Germany; nadine.strassberger-nerschbach@ukbonn.de (N.S.-N.); achilles.delis@ukbonn.de (A.D.); johannes.kamp@ukbonn.de (J.K.); alexandra.goertzen-patin@ukbonn.de (A.G.-P.); dishalen.cudian@ukbonn.de (D.C.); mark.coburn@ukbonn.de (M.C.); ehrenfried.schindler@ukbonn.de (E.S.); grigorij.schleifer@ukbonn.de (G.S.); maria.wittmann@ukbonn.de (M.W.); 2Department of Anesthesiology and Intensive Care Medicine, Hospital Vest, 45657 Recklinghausen, Germany; andreas.fleischer@klinikum-vest.de; 3Department of Anesthesiology, University of Groningen, University Medical Center Groningen, 9713 GZ Groningen, The Netherlands; j.k.g.wietasch@umcg.nl

**Keywords:** telemedicine, patient communication, remote informed consent, legal basis, European practice

## Abstract

Background: Digitalization in the health system is a topic that is rapidly gaining popularity, and not only because of the current pandemic. As in many areas of daily life, digitalization is becoming increasingly important in the medical field amid the exponential rise in the use of computers and smartphones. This opens up new possibilities for optimizing patient education in the context of anesthesia. The main aim of this study was to assess the implementation of remote consent in Europe. Methods: An online survey entitled “Digital online Patient Informed Consent for Anesthesia before Elective Surgery. Recent practice in Europe,” with a total of 27 questions, was sent by the European Society of Anesthesiology and Intensive Care (ESAIC) to their members in 47 European countries. To assess the effect of the economy on digitalization and legal status with regard to anesthesia consent, data were stratified based on gross domestic product per capita (GDPPC). Results: In total, 23.1% and 37.2% of the 930 participants indicated that it was possible to obtain consent online or via telephone, respectively. This observation was more often reported in countries with high GDPPC levels than in countries with low GDPPC levels. Furthermore, 27.3% of the responses for simple anesthesia, 18.7% of the responses for complex anesthesia, and 32.2% of the responses for repeated anesthesia indicated that remote consent was in accordance with the law, and this was especially prevalent in countries with high GDPPC. Concerning the timing of consent, patients were informed at least one day before in 67.1% of cases for simple procedures and in 85.2% of cases for complex procedures. Conclusion: Even European countries with high GDPPC use remote informed consent only in a minority of cases, and most of the time for repeated anesthetic procedures. This might reflect the inconsistent legal situation and inhomogeneous medical technical structures across Europe.

## 1. Introduction

In recent years, efforts have been made to shorten patients’ hospital stays and to streamline processes [1]. On the one hand, these efforts have an economic background, and on the other hand, there are medical advantages, because shorter hospital stays can, for example, lead to a reduction in nosocomial infections [2].

The World Health Organization (WHO) highlights healthcare-associated infections (HCAIs) as a major issue for healthcare providers, patients, and public authorities worldwide [3]. The European Centre for Disease Prevention and Control (ECDC) reported that approximately 4,131,000 patients are affected by 4,544,100 episodes of HCAI annually in Europe, with a mean HCAI prevalence of 7.1% [4]. In this context, HCAIs cause 16 million additional days of hospital stay and 37,000 attributable deaths (contributing to an additional 110,000). This amounts to approximately EUR 7 billion in associated costs annually (WHO, 2011), with considerable variability in estimates ranging from 1.2 to 26.4 excess days due to HCAI [5]. Additionally, fewer hospital visits would avoid unnecessary burden on the traffic system and have a beneficial impact on the environment.

Enabling remote contact and communication with patients using digital media tools instead of physical visits may facilitate a reduction in HCAIs. Especially for surgical interventions, this may help to keep the pre-operative hospital length of stay as short as possible. Currently, however, pre-operative anesthesiological evaluations are performed in most hospitals during the pre-operative stay or on an outpatient basis in advance. This practice is supported by the recommendations and guidelines of many medical societies that emphasize physical contact in the form of consultation, during which a patient is to be legitimately informed about his/her future anesthetic procedure [6].

Especially in light of the COVID-19 pandemic, a reduction in the length of hospital stay is of utmost desirability. Furthermore, social distancing is even more important for vulnerable patient groups, such as those with multiple comorbidities and immunosuppressed or elderly patients [7], even beyond the pandemic.

Despite all the advantages, it also remains to be mentioned that older people or those from lower income strata in particular do not always have access to computers or the Internet. In addition, in the future, there will still be a certain clientele of patients who, due to the severity and complexity of their illness, prefer a personal conversation with the attending physician.

The goal of this study was to identify differences in the implementation of telemedicine for pre-operative anesthesiological consultation across Europe. Furthermore, anesthesiologists’ knowledge of the legal framework was assessed, and the technical applicability of remote consent was highlighted.

## 2. Materials and Methods

To provide a Europe-wide overview of the current state of technical support and legal frameworks for telemedical consultation, a cross-sectional study in the form of a survey comprising 27 questions was sent through the European Society of Anesthesiology and Intensive Care (ESAIC) to its members in 47 European countries.

The survey was designed as a multiple choice questionnaire, with the possibility of adding comments or omitting questions. Written answers were not analyzed in this study and did not contribute to the results presented here. Test runs revealed that the questionnaire could be completed within 7–10 min. It was possible to resume the survey if the session had to be interrupted, and respondents were allowed to omit questions.

The main focus of the survey was particularly on the general and technical possibility of digital education in the field of anesthesia and the assessment of whether a remote online or telephone-based anesthesia consent procedure could comply with the current prevailing legal frameworks across different European countries. The participating anesthetists were asked to judge this according to their own assessment and to the best of their knowledge. Information was also collected on the timing of the preoperative visit, and a distinction was made between low- and high-risk procedures.

To analyze the effect of economic health on our results, a separate sub-analysis was performed. To accomplish this task, the implementation of Internet- and telephone-based solutions and doctors’ knowledge of legal requirements for remote consent were investigated in relation to the gross domestic product per capita (GDPPC). This was considered when discussing whether significant disparities existed or not. Quantitative data for GDPPC were accessed online on 16 February 2022 (https://data.worldbank.org/indicator/ny.GDPPC.pcap.cd), and the countries represented in our study were assigned to one of three groups: high, middle, and low GDPPC. The assignment process was performed such that each group had a similar number of countries. A detailed overview is provided in the Supplementary Material (Appendix A).

The survey was implemented using an open-source online questionnaire tool, “LimeSurvey CE” (Version 5.1, https://community.limesurvey.org/downloads/), and it was hosted on a secured Linux Debian (Version 10.11) server. The link to the questionnaire was distributed by the ESAIC communication committee (https://kai-survey.de/limesurvey/733779/) to 42,433 active members, and the survey was conducted over a three-week period (July to August 2021).

Absolute numbers and proportions of responses were calculated for the questions used in the survey. Differences in the study population and sub-analysis groups were compared using the chi-squared test for independence. The nature of the association between the row (GDPPC) and the column (the likelihood of giving a certain answer) in contingency tables was interpreted by identifying cells with the highest Pearson residuals as an estimate of the raw residuals’ standard deviation (r). The cells with the highest Pearson residuals show the direction and strength of the associative effect between the dependent and independent variables. Residuals that exceed 2 in absolute value indicate a strong association between gross domestic product per capita and the likelihood of giving a certain answer. The significance level was set at *p* ≤ 0.05, and all analyses were performed using the statistical language R (R Core Team, Vienna, Austria, Version 3.6.2). An interactive web application was programmed to provide additional insights into the survey population (https://kai-survey.shinyapps.io/ESAIC-KAI-survey-2021). All codes for statistical analysis and visualization can be accessed online (https://github.com/GrigorijSchleifer/EJA-ESAIC-survey).

## 3. Results

The survey “Digital Online Patient Informed Consent for Anesthesia before Elective Surgery—Recent Practice in Europe” conducted by the University Hospital Bonn and the ESAIC, was completed by 930 participants from 47 European countries. Overall, the data were provided by medical doctors (99%, *n* = 920), nurses (0.2%, *n* = 2), physician assistants (0.6%, *n* = 6), and other undisclosed professionals (0.2%, *n* = 2). Of the survey participants, 56% (*n* = 521) were male, 43.6% (*n* = 406) were female, and 0.3% (*n* = 3) were diverse. Consent was obtained predominantly by consultants (78.6%, *n* = 731) or residents (15.4%, *n* = 143) (Table 1). Most of the answers were contributed by colleagues from Germany (14.2%, *n* = 132), Spain (7.8%, *n* = 73), and Switzerland (7%, *n* = 65) (Table 2). Two participants did not provide professional- or expert-level information but were included in the overall analysis.

To assess the availability of digital media that facilitate remote informed consent across Europe, it was asked if it was possible to obtain consent via the Internet or telephone. In 70.2% (*n* = 486) of the responses, informed online consent was not possible or it was not uniformly implemented (6.6%, *n* = 46). However, 23.1% (*n* = 160) of the respondents stated that consent via the Internet was already used in clinical routines. Anesthesia consent via the telephone could not be obtained in 56.7% (*n* = 391) of cases. In contrast, 37.2% (*n* = 257) of the respondents answered that patient education via telephone was available, and 6.1% (*n* = 42) mentioned varying technical solutions (Figure 1). Based on this observation, consultation via telephone seems to be more frequently used for anesthesia informed consent.

To shed light on a possible association between economic health and the use of Internet- or telephone-based solutions for remote consent, data were stratified based on GDPPC. The strength and direction of the association between GDPPC and the use of the Internet or telephone was estimated by calculating Pearson residuals, as described in the statistics section. While in countries with low GDPPC, Internet usage seemed to be less common (Pearson residual = −1.18), countries with higher GDPPC were positively associated with online remote consent (Pearson residual = 1.5) per capita (Figure 2A). Interestingly, a remote consent process for anesthesia via the telephone was even more associated with increased GDPPC. The telephone was not commonly used for anesthesia consent in countries with low GDPPC (Pearson residual = −2.57), and its implementation was strongly associated with increased economic status (Pearson residual = 4.56) (Figure 2B).

Based on a previously published sub-analysis of the pediatric population, it was shown that legal regulations vary considerably across Europe [8]. Here, we assessed whether Internet or telephone consent was in accordance with the legal requirements for the adult patient population. For simple procedures, 37.2% (*n* = 298) of respondents stated that remote informed consent was not in accordance with legal regulations in their country, while it was legally sound in 27.3% (*n* = 219) of cases. For complex procedures, remote consent was not possible owing to legal requirements in 42.5% (*n* = 341) of cases, and only in 18.7% (*n* = 150) of cases was it compliant with the law according to the participants’ understanding of their judiciary setup. Overall, respondents were unsure about legal regulations with respect to Internet or telephone informed consent 35.5% (*n* = 285) of the time for simple procedures and 38.8% (*n* = 311) of the time for complex procedures (Figure 3).

To estimate the effect of economic status and digitalization on the legal framework with respect to remote consent for simple, complex, and repeated anesthesia, data were stratified based on GDPPC, as described in the Methods section. The data showed a high positive association between high GDPPC and remote informed consent in accordance with the legal requirements for simple, complex, and repeated anesthesia (Pearson residuals: 5.37 (simple), 4.33 (complex), and 5.35 (repeated)). On the contrary, in countries with low GDPPC, online remote consent was more often stated as not being in accordance with legal requirements for simple, complex, and repeated anesthesia (Pearson residuals: −4.17 (simple), −2.98 (complex), and −4.21 (repeated)) (Figure 4). These observations suggest that with improved economic status, legal regulations tend to favor remote consent processes, whereas in countries with weaker economies, remote anesthesia processes are discouraged.

It was further investigated whether procedural severity affected the timing of obtaining informed consent. For simple procedures, informed consent was obtained 12.9% (*n* = 103) of the time 2 days or more before surgery, and in 54.2% (*n* = 433) of cases, it was obtained 24 h or less before surgery (Figure 5). In 32.9% (*n* = 263) of cases, consent was obtained on the same day of surgery. In comparison, for complex procedures, consent was obtained in 33.6% (*n* = 269) of cases 2 days or more before surgery, and 51.6% (*n* = 413) of the time it was obtained 24 h or less before the surgical intervention. On the day of surgery, consent was acquired in only 14.8% (*n* = 118) of cases. Here, we observed that patients who needed an extended procedure were more likely to stay overnight than those who underwent simple procedures.

The effect of economic status (GDPPC) on anesthesiologist preference to obtain informed consent in the future was also assessed. In all three GDPPC groups, most of the respondents answered that they would still prefer to obtain informed consent in person (high GDPPC: 75.1% (*n* = 269), middle GDPPC: 83.6% (*n* = 102), low GDPPC: 79.1% (*n* = 216), Table 3). Remote anesthesia consultation was mentioned less frequently, and it was independent of GDPPC grouping (high GDPPC: 24.9% (*n* = 89), middle GDPPC: 16.5% (*n* = 20), low GDPPC: 20.8% (*n* = 57), for detailed information see Table 4). Overall, the associative effect between GDPPC and preference was low.

## 4. Discussion

This study showed that the implementation of telemedical support in the field of anesthesiological pre-operative consultation varied considerably across Europe. Remote informed consent was used only in a minority of cases, and there was widespread uncertainty regarding legal frameworks. Additionally, the rate of remote pre-operative consultations strongly correlated with the economic status of the European countries surveyed.

### 4.1. Technical and Practical Implementation

Telemedical support in the field of medicine has been available for years, but thus far, it has only been used with restraint [9]. The obvious advantages, such as reduction of waiting times and avoidance of unnecessary travel [10], combined with an increasing desire for social distancing during the COVID-19 pandemic, may have given impetus to implementing telemedicine tools for remote consultation. However, our survey showed that it was not possible for most respondents to obtain anesthesiological consent via the Internet or telephone. This observation was especially prevalent in European countries with low gross domestic product per capita (GDPPC).

It is well realized that access to the Internet is a social determinant of health [11,12], and that telemedicine can improve patient safety [13]. Nevertheless, equal access to resources that enable telemedical patient consultation requires affordability for healthcare institutions [14], which is not evenly spread across Europe. Furthermore, the requirement for written consent also poses problems in practice, as electronic signature technology cannot be implemented everywhere and has not yet become routine in the health sector, in contrast to its increasingly widespread use in other areas of daily life [15].

Thus, an improvement in technical standards in poorer countries could lead to the wider use of digital media and thus benefit certain patient groups [16].

It remains to be mentioned that especially in the sensitive area of healthcare, data security must have top priority. Therefore, when using remote technology, it is essential to ensure that the corresponding devices and data transfers comply with data protection guidelines [17].

### 4.2. Legal Basis of Informed Consent

More than 35% of the respondents were unsure about the legal framework conditions for remote informed consent, and up to 42.5% feared that it was illegal depending on the procedure (Figure 3). This emphasizes the necessity of a clear legal framework within Europe that could be beneficial for patients and involved anesthesiologists.

The overall assessment of the legal situation shows a clear picture in our survey. Only 19 to 32% percent of participants believed that telemedical information is legal, whereas 33 to 43% answered that it is not in accordance with the legal requirements. As already mentioned, a very large proportion of respondents were unclear about the legal regulations (35–39%). The restrictive attitude may be due to the fact that there is no clear legislation in this regard and the anesthetist is legally on thin ice when it comes to the use of telemedicine media. Healthcare is the responsibility of individual member states, and there is no uniform, Europe-wide regulatory body. Specific legal regulations for the use and handling of telemedicine are lacking in many countries, and harmonization across the EU is often described as unfeasible because of data protection regulations [9]. In 2020, Nittari et al. published a review about the current ethical and legal challenges in telemedicine practice. Here, the most important points, which were noted several times in the various works, were the need for a European or international regulation that adapts to the evolving technologies, and the need for a clear reimbursement policy for telemedical services [18].

Currently, a new electronic cross-border health service is being established for the EU (MyHealth@EU) [19], and the European Union is taking a first but giant step toward the unification of data exchange systems via the creation of the European Data Health Space [20]. The step to create laws at the European level that regulate the handling of telemedicine is a sensible path, even if implementation in all countries of the EU will take time.

Comparing countries with a higher GDPPC with those with a lower GDPPC, we find that in the first group, remote informed consent is more often considered legitimate (Figure 4). Therefore, regulations for all EU countries remain a promising future prospect that could benefit the healthcare system as a whole and many patients.

### 4.3. Time Frame of Informed Consent

Looking at the time frame, in most cases, the patient was informed at least one day before the procedure; this applied to simple and complex procedures (67.1% vs. 85.2%).

The current procedure for major surgery is an inpatient stay where the patient’s consent is usually obtained within 24 to 48 h, allowing for sufficient reflection time [21,22].

However, a time course of 24 to 48 h is often not sufficient to achieve comprehensive pre-operative risk mitigation [23]. In order to have adequate lead time for the potential pre-operative improvement of the patient’s health, one approach might be to provide information via the Internet during video consultation days to weeks before their hospital stay. This might be a systematic attempt to improve patient safety, which should be the primary goal [23]. Another benefit might be the avoidance of unnecessary pre-anesthesia on-site visits for patients without major pre-existing conditions. This could reduce costs, possibly save the patient a long journey, and reduce the risk of nosocomial infections. A similar study on the topic investigated the satisfaction of combined telephone and remote education in children compared to on-site consultation. Here, the first option was clearly favored [24].

In another study, it was shown that remote information for patients was not inferior to face-to-face conversation. Even though this was information about a clinical trial, it is evident that the content can be communicated equally well to the patients [25].

However, despite the potential benefits of telemedicine, especially in terms of less stress and shorter waiting times, the primary wish for the future in countries with high, medium, or low GDPPC is still to carry out face-to-face patient information (Table 3). The assessment in this regard may also be due to the fact that the legitimacy of purely remote patient information is often still doubted. However, in the age of increasing digitization, it should be legally possible to put online medical consultations on an equal footing with face-to-face consultations. In cases of doubt, the attending physician can decide with the patient which course of action makes the most sense. In addition, the patient should always be offered personal contact if desired, as older people or those with lower levels of education often have less access to digital media. In conclusion, legal pitfalls should not be unnecessary obstacles when communication via computers or smartphones has become a standard in many other areas of daily life.

### 4.4. Strength and Limitations

The questionnaire was sent to 42,433 active ESAIC members from 47 countries. The response rate was only 2.2%, with 930 responses, and responses were not equally distributed across all countries. Nevertheless, responses from anesthesiologists from 47 different nations were obtained.

Furthermore, although the questionnaire was designed with maximum care by experienced clinical and research anesthesiologists, a response bias may have arisen. As the questionnaire was only sent out once by the ESAIC, there was no possibility of pre-testing to evaluate the possible bias effect of the primer questions. Thus, the validity of the results could be limited because of question order effects. However, the survey was performed anonymously without any human contact, and socially desirable responses and interviewer bias should not be a concern.

## 5. Conclusions

The present survey was able to show that, especially in low income countries, telemedicine in the field of anesthesia is not frequently used in Europe, even though many professional societies have been recommending it for years [26,27,28]. There is obviously much uncertainty regarding the legal framework conditions. Hence, the authors conclude that a Europe-wide harmonization of laws concerning remote informed consent is desirable, even if it would be difficult to implement equally for all countries in the near future.

## Figures and Tables

**Figure 1 healthcare-11-01942-f001:**
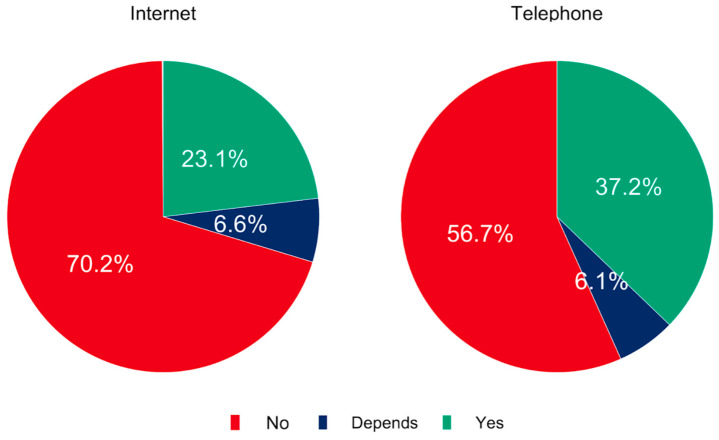
Is it possible to obtain informed consent online via the Internet or telephone in your routine setting?

**Figure 2 healthcare-11-01942-f002:**
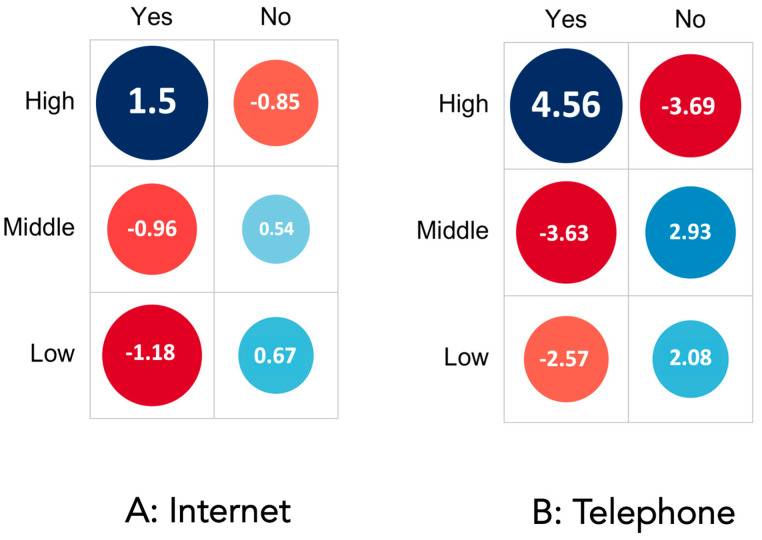
Is it possible to obtain informed consent online via the Internet (**A**) or telephone (**B**) in your routine setting? (Stratified by high, middle, and low gross domestic product per capita (GDP per capita)). Pearson residuals are used to represent the strength and direction of the association between GDP per capita and the response variable. In this graphical representation, a stronger positive association is visually depicted as a darker shade of blue, while a stronger negative association is represented by a darker shade of red.

**Figure 3 healthcare-11-01942-f003:**
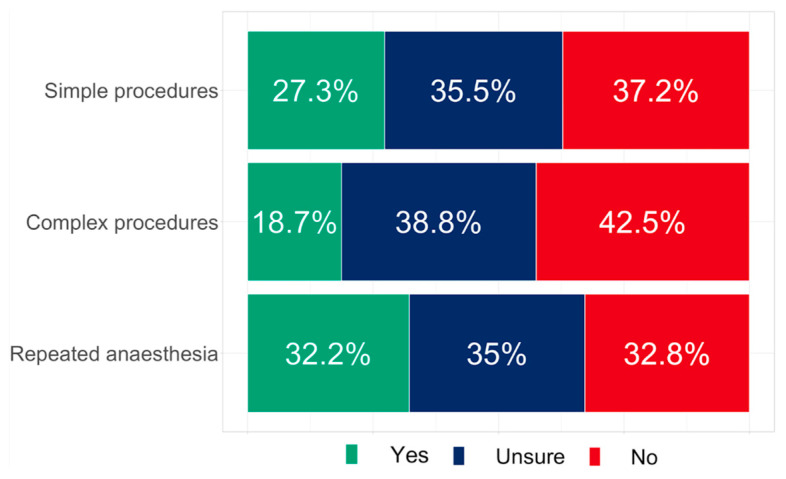
Is remote informed consent in accordance with the legal requirements for simple or complex procedures, and would it be allowed for repeated anesthesia?

**Figure 4 healthcare-11-01942-f004:**
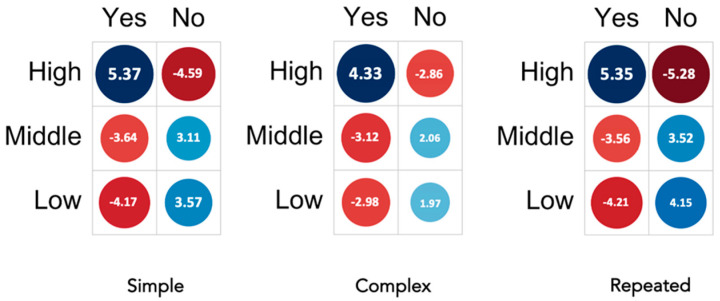
Is remote informed consent in accordance with the legal requirements for simple or complex procedures, and would it be allowed for repeated anesthesia? (Stratified by high, middle, and low gross domestic product per capita (GDP per capita)). Pearson residuals are used to represent the strength and direction of the association between GDP per capita and the response variable. In this graphical representation, a stronger positive association is visually depicted as a darker shade of blue, while a stronger negative association is represented by a darker shade of red.

**Figure 5 healthcare-11-01942-f005:**
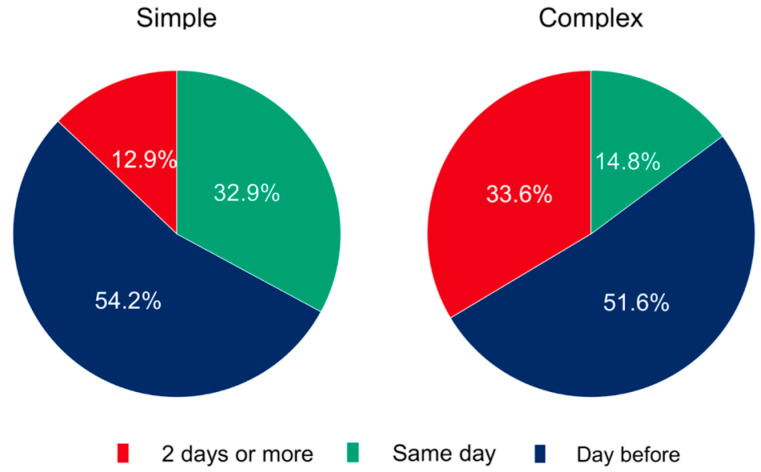
When do you need to obtain informed consent for elective surgery based on legal requirements for simple and complex procedures?

**Table 1 healthcare-11-01942-t001:** Descriptive statistics of the study population stratified by gender.

*n* (%)	Female	Male	Diverse	*p*
	406 (43.6)	521 (56)	3 (0.3)	
**Profession (%)**				<0.001
Medical doctor	402 (99.0)	516 (99.2)	2 (66.7)	
Nurse	0 (0.0)	1 (0.2)	1 (33.3)	
Physician assistant	4 (1.0)	2 (0.4)	0 (0.0)	
Other	0 (0.0)	1 (0.2)	0 (0.0)	
**Your expert level (%)**				0.33
Anaesthesia technician	4 (1.0)	6 (1.2)	0 (0.0)	
Consultant	303 (74.8)	426 (81.8)	2 (66.7)	
Resident	75 (18.5)	67 (12.9)	1 (33.3)	
Special trained nurse	0 (0.0)	1 (0.2)	0 (0.0)	
Other	23 (5.7)	21 (4.0)	0 (0.0)	

**Table 2 healthcare-11-01942-t002:** Number of responses from countries where colleagues participated in our survey.

Country of Employment? n (%)			
Albania	3 (0.3)	Liechtenstein	1 (0.1)
Austria	32 (3.4)	Lithuania	8 (0.9)
Belarus	2 (0.2)	Luxembourg	4 (0.4)
Belgium	27 (2.9)	Malta	6 (0.6)
Bosnia and Herzegovina	5 (0.5)	Moldova	3 (0.3)
Bulgaria	8 (0.9)	Monaco	1 (0.1)
Croatia	26 (2.8)	Netherlands	50 (5.4)
Cyprus	4 (0.4)	Macedonia	3 (0.3)
Czechia	10 (1.1)	Norway	8 (0.9)
Denmark	11 (1.2)	Poland	19 (2.0)
Estonia	4 (0.4)	Portugal	56 (6.0)
Finland	12 (1.3)	Romania	31 (3.3)
France	24 (2.6)	Russia	12 (1.3)
Georgia	2 (0.2)	Serbia	14 (1.5)
Germany	132 (14.2)	Slovakia	8 (0.9)
Greece	45 (4.8)	Slovenia	14 (1.5)
Hungary	8 (0.9)	Spain	73 (7.8)
Iceland	1 (0.1)	Sweden	43 (4.6)
Ireland	18 (1.9)	Switzerland	65 (7.0)
Israel	6 (0.6)	Turkey	23 (2.5)
Italy	40 (4.3)	Ukraine	7 (0.8)
Kazakhstan	2 (0.2)	United Kingdom (UK)	44 (4.7)
Kosovo	3 (0.3)	Uzbekistan	2 (0.2)
Latvia	10 (1.1)		

**Table 3 healthcare-11-01942-t003:** How would you prefer to conduct the anesthesiological preoperative evaluation in the future?

GDPPC	High	Middle	Low	
*n*	454	144	329	*p*
**Preference *n*** **(%)**				0.039
In person	269 (75.1)	102 (83.6)	216 (79.1)	
Self assessment online	39 (10.9)	3 (2.5)	14 (5.1)	
Video Conference online	40 (11.2)	13 (10.7)	35 (12.8)	
Telephone	10 (2.8)	4 (3.3)	8 (2.9)	

**Table 4 healthcare-11-01942-t004:** Contingency tables of the survey questions containing observed cell totals, expected cell totals (in round brackets) and residuals (in square brackets) stratified by high, middle and low Gross Domestic Product per capita (GDPPCPC)).

**Is it possible to obtain informed consent online via *internet* in your routine setting?**			***p* = 0.101**
	**Yes**	**No**	
**High GDPPCPC**	89 (75) [1.5]	222 (235) [−0.85]	
**Middle GDPPCPC**	49 (56) [−0.96]	181 (174) [0.54]	
**Low GDPPCPC**	19 (25) [−1.18]	83 (77) [0.67]	
**Is it possible to obtain informed consent online via *telephone* in your routine setting?**			***p* = <0.001**
	**Yes**	**No**	
**High GDPPCPC**	175 (124) [4.56]	139 (190) [−3.69]	
**Middle GDPPCPC**	56 (91) [−3.63]	173 (138) [2.93]	
**Low GDPPCPC**	24 (40) [−2.57]	78 (61) [2.08]	
**Is an online telephone informed consent for elective surgery in accordance with the** **legal requirements in your country (*simple*)**			** *p* ** ** = <0.001**
	**Yes**	**No**	
**High GDPPCPC**	164 (108) [5.37]	92 (148) [−4.59]	
**Middle GDPPCPC**	45 (77) [−3.64]	137 (105) [3.11]	
**Low GDPPCPC**	9 (32) [−4.17]	69 (45) [3.57]	
**Is an online telephone informed consent for elective surgery in accordance with the** **legal requirements in your country (*complex*)**			** *p* ** ** = <0.001**
	**Yes**	**No**	
**High GDPPCPC**	110 (73) [4.33]	130 (167) [−2.86]	
**Middle GDPPCPC**	30 (53) [−3.12]	143 (120) [2.06]	
**Low GDPPCPC**	9 (23) [−2.98]	68 (54) [1.97]	
**Thinking of repeated anaesthesia, would an online telephone informed consent** **then be allowed for elective surgery due to legal requirements?**			** *p* ** ** = <0.001**
	**Yes**	**No**	
**High GDPPCPC**	187 (127) [5.35]	70 (130) [−5.28]	
**Middle GDPPCPC**	54 (87) [−3.56]	123 (90) [3.52]	
**Low GDPPCPC**	14 (40) [−4.21]	69 (42) [4.15]	

## Data Availability

The data presented in this study are available in https://github.com/GrigorijSchleifer/EJA-ESAIC-survey.

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
