# Peer review of "Digital Online Patient Informed Consent for Anesthesia before Elective Surgery—Recent Practice in Europe"

_healthcare, 2023, doi:10.3390/healthcare11131942_

Round 1

Reviewer 1 Report

this article has an interesting subjects. However, the github  link address in article  is wrong and old, so access to github link through this link is impossible. in addition to, codes in github based on old address and can not reachable. please write the correct and update links of github in article and new access link in tab of  cods at githab.

In the discussion, it seems good if some similar researches are mentioned and the results are compared with other studies.

Author Response

Reviewer 1

Comments and Suggestions for Authors

this article has an interesting subjects. However, the github  link address in article  is wrong and old, so access to github link through this link is impossible. in addition to, codes in github based on old address and can not reachable. please write the correct and update links of github in article and new access link in tab of  cods at githab.

We thank the reviewer for this important information. The link has been updated.

In the discussion, it seems good if some similar researches are mentioned and the results are compared with other studies.

According to the author's request, two more studies on the topic of remote education were cited in the discussion section, but the topic has not yet been intensively investigated across Europe in this respect. (page 11 / lines 282-286, page 12/ lines 311-316)

Reviewer 2 Report

1) The Authors performed an article entitled "Remote anaesthesia patient informed consent - an overview of the current implementation in Europe". Overall, the publication is interesting and is well organized and written. However, some aspects should be improved and commented.

2) The microbial contamination of computers and telephones in hospitals/health spaces is described in some studies. Please comment this aspect. In my opinion, this information should be discussed in the article.

3) In general, the elderly and people with low education/literacy do not have good digital/internet/telephone skills. Please comment this aspect. In my opinion, this information should be discussed in the article.

4) The response rate was very low (only 2.2 %). Why did not you extend the time to obtain answers from the questionnaires? Please comment this information.

5) Suggestion: Please do not use "we" in the publication. I think it is better to write in a formal way.

Author Response

Reviewer 2

Comments and Suggestions for Authors

1) The Authors performed an article entitled "Remote anaesthesia patient informed consent - an overview of the current implementation in Europe". Overall, the publication is interesting and is well organized and written. However, some aspects should be improved and commented.

We thank the reviewer for the kind assessment of our article and have worked on the suggestions point by point.

2) The microbial contamination of computers and telephones in hospitals/health spaces is described in some studies. Please comment this aspect. In my opinion, this information should be discussed in the article.

The reviewer raises an important point. A corresponding paragraph with literature reference has been added under 4.1 Technical and practical implementation. (page 11/ lines 264-267)

3) In general, the elderly and people with low education/literacy do not have good digital/internet/telephone skills. Please comment this aspect. In my opinion, this information should be discussed in the article.

Thank you for this important point. In the discussion section was added:

“as older people or those with lower levels of education often have less access to digital media.” (page 12 /lines 325-326)

4) The response rate was very low (only 2.2 %). Why did not you extend the time to obtain answers from the questionnaires? Please comment this information.

ESAIC sends out questionnaires on topics of interest several times a year, which go through a selection process beforehand. Here, it is in line with the usual procedure that the questionnaires are only sent to the members once and can also only be processed online for a certain period of time. The authors had no influence on this. This was also presented under Limitations.

5) Suggestion: Please do not use "we" in the publication. I think it is better to write in a formal way.

The language correction was made and the corresponding changes were highlighted in color.

Reviewer 3 Report

The manuscript on remote anesthesia patient informed consent is well-written.

However, one comment should be considered before acceptance: the questionnaire used in the study was not sufficiently explained, including its validity and reliability assessment.

Author Response

Reviewer 3

Comments and Suggestions for Authors

The manuscript on remote anesthesia patient informed consent is well-written.

However, one comment should be considered before acceptance: the questionnaire used in the study was not sufficiently explained, including its validity and reliability assessment.

We thank the reviewer for the positive evaluation of our article.

As shown in section 4.4 Strength and Limitations, no specifically validated questionnaire was used. All of the anesthesiologists involved in the project have years of professional experience and an in-depth research background, and thus expertise in creating questionnaires to collect data on specific questions. It was explained why corresponding biases are unlikely and, moreover, the survey was extensively reviewed by the Editorial-board of ESAIC before publication.

Thus, we hope to have allayed the reviewer´s concerns in this regard.

Reviewer 4 Report

Dear Authors

This is an interesting and useful piece of research, and I would recommend publication. However, it needs a bit more detail, either in the text or in longer footnotes.

Yours faithfully

Reviewer

Author Response

Reviewer 4

Comments and Suggestions for Authors

Dear Authors

This is an interesting and useful piece of research, and I would recommend publication. However, it needs a bit more detail, either in the text or in longer footnotes.

Yours faithfully

Reviewer

We thank the reviewer for the positive assessment and the recommendation for publication.

Title: “Remote anaesthesia patient informed consent - an overview of the current implementation in Europe” *Please correct the error in the title of the article (anaesthesia patient).

Language correction into American English has been made throughout the text.

Reviewer Comments

This is an interesting and useful piece of research, and I would recommend publication. However, it needs a bit more detail, either in the text or in longer footnotes.

General points: Digitalization in the healthcare system is an important and current topic, which is rapidly gaining popularity. In every area of human life not only privately but also professionally, the use of computers and smartphones is a helpful tool for gaining knowledge. In medicine the use of digitalization offers new possibilities, which accelerate a process of contact between a patients and physicians, receiving the information about the diagnosis and proposes treatment methods in a short time.

Authors of the paper decided a put their attention on a possibilities of implementing specific procedures that would enable remotely obtaining an informed consent from anesthetic patients before the planned surgery. This study was carried out using an original 27-question a survey that was sent to the members of the European Society of Anesthesiology and Intensive Care (ESAIC) from 47 European countries. In the introduction of this paper, the authors explain, that digitalization in an healthcare system reduces the number of hospital visits which has positive impact on the economical (avoiding unnecessary burden on the healthcare system) and medical (reduced number of healthcare-associated infections) aspects.

Are these benefits the only ones, and does telemedicine not also have some limitations that should be mentioned in the introduction to the paper? Please explain this issue in detail.

We thank the reviewer for this important remark and have added a corresponding paragraph in the introductory section. (page 2 /  lines 63-67)

In the research part, the authors estimated the impact of economic status and digitization on the legal framework for obtaining informed consent from anesthesia patients. I am asking for a detailed explanation of the terms of simple, complex and repeated procedures with a reference to the legal aspects of countries with different economic status.

We thank the reviewer for this comment. According to research, there is a very wide spectrum of legal aspects within Europe. While in some countries face-to-face information of the patient is obligatory, in Germany, for example, it is permitted to obtain consent for anaesthesia purely by telephone in "simple" procedures that are not precisely defined by the legislator.

Therefore, in this survey, it was left to the anaesthetist to assess the procedure as simple or complex, or how they would proceed with a repeat anaesthesia.

In other words, will diagnostic tests and invasive procedures that are complex procedures in high-status countries have a similar legal justification for their use in other low-status countries?

The reviewer raises an interesting point. However, an analysis of this would go beyond the focus of this paper. In addition, the legal aspects are very complex and, due to the increasing use of technology, are also in a state of flux, making this question difficult to answer.

That is why the authors wanted to advocate harmonisation throughout Europe.

Specific points: L 114 - in the Materials and Methods section, please provide a valid website address, because the one below is not active. https://github.com/GrigorijSchleifer/EJA-ESAIC-survey Comments on Discussion (Legal basis of informed consent section)

As noted by another reviewer, the link is obviously no longer accessible and we apologise for this. A new functioning link has been added to the paper.

Legal regulations in different European countries differ in terms of obtaining remotely informed consent. Please briefly characterize them, emphasizing the most important aspects that distinguish countries with a high, medium and low economic status and comparing them with the results presented by the authors of the study.

As already mentioned above, the legal regulations in Europe are so different that a detailed description would go beyond the scope of this paper. In terms of legality, for example, there are serious differences even within countries with a high economic status.

However, since the authors have had the same idea of taking a closer look at the legal aspects within Europe, an article is indeed planned that will deal precisely with this topic.

Comments on conclusions Please make some suggestions to improve technical standards in poorer countries that would lead to a wider use of digital media, offering benefits to certain groups of patients

Improving technical standards in poorer countries is more of a political issue and here anesthesiologists alone will achieve little. However, if the reviewer recommends it, the conclusion could end with the following sentence:

This might also be an appeal to the politicians of the better-off countries to support the poorer countries of Europe financially with regard to the important issue of further advancing digitalisation.
